# Current and Novel Alkylators in Multiple Myeloma

**DOI:** 10.3390/cancers13102465

**Published:** 2021-05-18

**Authors:** Fredrik Schjesvold, Albert Oriol

**Affiliations:** 1Oslo Myeloma Center, Oslo University Hospital, 0450 Oslo, Norway; 2K.G. Jebsen Centre for B-Cell Malignancies, University of Oslo, 4950 Oslo, Norway; 3Institut Josep Carreras and Institut Català d’Oncologia, Hospital Germans Trias I Pujol, 08916 Badalona, Spain; aoriol@iconcologia.net

**Keywords:** alkylator, myeloma, melflufen, melphalan, bendamustine, cyclophosphamide

## Abstract

**Simple Summary:**

In this review we have summarized the history, the current use, and the future possibilities of alkylator treatment in multiple myeloma. Alkylators have for decades been part of the standard of care of myeloma treatment, but still new alkylators and new use of old alkylators are making its way into myeloma guidelines of today.

**Abstract:**

A large number of novel treatments for myeloma have been developed and approved; however, alkylating drugs continue to be part of standard regimens. Additionally, novel alkylators are currently being developed. We performed a non-systematized literary search for relevant papers and communications at large conferences, as well as exploiting the authors’ knowledge of the field, to review the history, current use and novel concepts around the traditional alkylators cyclophosphamide, bendamustine and melphalan and current data on the newly developed pro-drug melflufen. Even in the era of targeted treatment and personalized medicine, alkylating drugs continue to be part of the standard-of-care in myeloma, and new alkylators are coming to the market.

## 1. Introduction

Before alkylators, no drugs could alleviate the sufferings of multiple myeloma (MM) patients, but not for lack of trying. Leeches, steel, quinine, urethane rhubarb and orange peel were all tried, but not to much avail [1]. Alkylators constituted the first effective weapon in myeloma treatment and its use evolved through being the mainstay of handling myeloma patients for decades, to today, being an agent to combine with several other agents with different mechanisms of action in the armamentarium of a myeloma doctor.

Alkylators, in the form of “mustard gas”, were first used by the German army in World War I, and, already in 1919, they were known to be suppressors of hematopoiesis [2], leading to mustine being the first chemotherapy used for patients with different forms of leukemia and lymphoma [3]. The drugs’ mechanism is to add an alkyl group to DNA, causing linking between the two strands, inhibiting DNA and RNA synthesis and, thus, leading to cell death, preferentially in highly proliferative cells (Figure 1).

Melphalan was first synthesized in 1953 as a derivative of nitrogen mustard [4], eventually leading to the first documentation of clinical benefit in 3 of 6 MM patients [5]. A decade later, the benefits of the combination of melphalan with prednisolone (MP) were demonstrated in a fairly large, randomized trial of 183 participants [6], establishing MP as a standard regimen for the rest of the century. The regimen is still in use in combination with other drugs.

The next big leap in myeloma treatment was to use high doses of melphalan, in the context of autologous stem cell transplant. After the first positive report in 1983 [7], several studies accumulated on the benefit of what became known as high-dose-melphalan with autologous stem cell transplant (HDM-ASCT), and, after a large, randomized study published in 1996 [8], it became established as the standard-of-care for young newly diagnosed myeloma patients. Over the years, several other alkylating drugs and other chemotherapeutics have been tested and used, including cyclophosphamide, carmustine, lomustine, bendamustine, cisplatin, vincristine and doxorubicin, also in the context of allogeneic stem cell transplantation [9]. One of the more noteworthy attempts was the VTD-PACE (bortezomib, thalidomide, dexamethasone, cisplatin, doxorubicin, cyclophosphamide and etoposide) regimen for induction and consolidation before transplant, in the total therapy 3 study [10], showing near-complete remission and 2-year survival rates in more than 80% of the patients. The regimen is toxic, but VTD-PACE and similar regimens are still being used occasionally for refractory patients at some centers [11].

## 2. Current Use of “Old” Alkylating Agents

### 2.1. Current Combinations with Melphalan

Therapy with melphalan plus prednisone (MP) was the standard of care for patients with newly diagnosed MM for almost 40 years [6,12] and when alternative combinations were developed as induction therapy for individuals eligible for high-dose melphalan therapy with hematopoietic stem-cell transplantation [8,13], it remained the standard of care for elderly or unfit patients. Several combinations have built over the MP backbone. Preclinical studies proved that the proteasome inhibitor bortezomib sensitized melphalan-sensitive and melphalan-resistant myeloma cell lines to melphalan and down-regulated cellular responses to genotoxic stress [14,15]. The combination of bortezomib with MP was the first approved indication for bortezomib in the MM front-line setting [16]. Long-term follow-up of patients treated with MP plus bortezomib (MPV) showed unprecedented progression-free (PFS) and overall survival (OS) rates in newly diagnosed non-transplant candidate MM patients, with median durations of almost 2 and 5 years, respectively [17,18]. In the same period, the combination of thalidomide plus MP (MPT) was also compared with MP in six clinical trials [19,20,21,22,23], leading to a meta-analysis showing clear benefit and, thereafter, to the approval of the MPT regimen in the front-line setting [24]. PFS was 20.3 months and OS was 39.3 months. However, MPT proved inferior to the more convenient alkylator-free combination of lenalidomide and dexamethasone (Ld) [25] and, today, it is not used when other options are available. Other MP combinations have also been tried. MP plus lenalidomide was associated with a high rate of severe neutropenia [26] and did not provide a clear benefit over Ld [27] or MPT [28], and MP plus carfilzomib could not prove any significant benefit over MP plus bortezomib [29].

Standard dose melphalan, in combination, is typically used in a limited duration schedule up to 12 to 18 months. Subsequent evidence favored continuous rather than limited duration therapy [30], but the original MPV schedule using bi-weekly intravenous bortezomib was associated with a high incidence of peripheral neuropathy, leading to early bortezomib discontinuation in up to 30% of patients [16]. Several trials [31,32] successfully explored less intensive initial bortezomib dosing to reduce toxicity and to allow some modality of bortezomib maintenance; however, such dose modifications were not associated with major improvements in terms of efficacy over the original MPV schedules. Finally, the phase 3 randomized trial (ALCYONE) compared 12 months of MPV alone with MPV plus daratumumab (D-VMP) for 12 months followed by daratumumab maintenance in patients with newly diagnosed myeloma who were ineligible for high-dose therapy [33]. Its first prespecified interim analysis, after a median follow-up of 16.5 months, showed an 18-month PFS rate of 71.6% in the daratumumab group, in contrast with 50.2% in the control group. The hazard ratio for disease progression or death was 0.50 (95% confidence interval 0.38 to 0.65; *p* < 0.001). The benefit of the D-VMP combination was also observed in terms of overall response rate (90.9% in the daratumumab group, as compared with 73.9% in the control group; *p* < 0.001), complete response or better (42.6% versus 24.4%; *p* < 0.001) and in terms of negativity for minimal residual disease (22.3% as compared with 6.2% at a threshold of 1 tumor cell per 105 white cells; *p* < 0.001). A recently published update, at a median follow-up of 40 months, confirmed the PFS benefit for the D-VMP group (hazard ratio 0.42 [0.34–0.51]; *p* < 0.0001), and also observed a significant benefit in OS with a hazard ratio for death of 0.60 (95% CI 0.46–0.80; *p* = 0.0003). The Kaplan–Meier estimate of the 36-month rate of OS was 78% (95% CI 73–82) in the D-VMP group and 68% (63–73) in the VMP group [34]. Despite a higher rate of grade 3 or 4 infections (23% in the daratumumab group and 15% in the control group) and the occurrence of daratumumab-associated infusion-related reactions in 27.7% of the patients, the overall safety profile of D-VMP was adequate and led to the approval of the combination for newly diagnosed MM patients who are not candidates to autologous SCT. D-VMP was the first approved combination that included daratumumab in this setting, and also the first approved treatment strategy that included daratumumab as maintenance therapy.

### 2.2. Current Status of HDM

After a large, randomized study comparing conventional chemotherapy and HDM-ASCT, transplant became the standard-of-care for patients fit enough to go through the procedure [8]. In this study, the conditioning was a combination of melphalan (140 mg per square meter) and total-body irradiation (4 × 8Gy). A later study demonstrated the benefit, both in terms of safety and OS, of a higher dose of melphalan (200 mg per square meter) without total-body irradiation [35], and this has since remained the standard dose in this setting [36]. One recent randomized trial has challenged HDM-ASCT following currently preferred induction treatment with both an immunomodulatory agent and a proteasome inhibitor [37]. The IFM2009 study included 700 patients randomized to either eight cycles of RVd (Revlimid, Velcade, dexamethasone) or three induction cycles, HDM-ASCT and two additional consolidation cycles. Lenalidomide maintenance was given in both arms for 13 cycles. PFS was increased in the transplant-arm (47.3 m vs. 35.0 m; *p* = 0.0001), but even after extended follow-up [38], no differences in PFS2 or OS have been observed. However, 201 patients in the non-transplant arm received transplant at relapse. In this context, the NCCN guidelines, despite calling ASCT the “preferred option”, consider delayed transplant (at relapse) “appropriate as well” [39]. The use of HDM-ASCT, however, is still increasing in all regions of the world. In North America, the percentage of patients receiving HDM-ASCT, in 2015, was the highest in the world, and the highest ever (25.70%). Among patients under the age of 70, the utilization rate in North America was 52.17% [40]. In Europe, the respective numbers were 22.24% and 46.68%, also the highest ever. In the rest of the world, the usage is lower, but increasing in all regions, with a global utilization rate in 2015 of 15.40%. Although it has been argued that a MRD-driven approach, only forwarding MRD-positive patients after induction regimen to transplant [41], may reduce toxicity without compromising efficacy, such an approach has not been sustained in data from clinical trials so far.

A second transplant has been used in many countries as part of standard-of-care, either at first relapse, or as a consolidation of the first transplant, often referred to as tandem transplantation. Three recent first-line trials, carried out in Europe, performed single or tandem transplantation according to their local standards of care; while Italy and Germany performed tandem transplantation, patients in Spain and the Netherlands received a single procedure. A meta-analysis, so far unpublished, of these trials argued in favor of tandem transplantation [42]. In all, 909 patients were included in the analysis, 501 with single and 408 with tandem transplantation. OS at 120 months was 55% and 42% for tandem and single transplantation, respectively (HR 0.69; *p* < 0.001). Similar results were achieved in the EMN02/HOVON95 study, in which a sub-study randomized patients after four induction cycles of bortezomib, cyclophosphamide and dexamethasone (VCD) to single or double transplant [43]. In this trial, there was also a significant benefit in terms of OS (HR 0.62; *p* = 0.022) in the intention-to-treat population. Based on this, the ESMO guidelines recommend tandem transplantation to all patients who have received VCD as induction [36]. Another study (BMT CTN 0702) also randomized patients to single or tandem transplant but did not show any benefit regarding neither PFS nor OS [44]. In this study, patients had received different forms of induction treatment within a 4–12-month period, which makes the population less homogenous. In addition, 32% of the patients allocated to tandem transplant did not receive this treatment. A post-hoc analysis with longer follow up demonstrated a benefit in 6-year PFS in the as-treated group with tandem transplant vs. single transplant with 49.4% vs. 39.7%; *p* = 0.01 [45]. In both studies, the benefit was more pronounced in the high-risk population. Current ESMO guidelines propose tandem transplantation in patients with high-risk cytogenetic features [36], and the NCCN guidelines state that it should be considered for all patients who are candidates for HDM-SCT [39].

If a transplant is performed front-line, a second transplant at relapse can be considered, and has been commonly used in many countries. Two randomized trials have been performed to evaluate transplant at first relapse. The British Myeloma X trial randomized patients after induction with bortezomib, doxorubicin and dexamethasone (PAD) to a salvage transplant or to weekly oral cyclophosphamide for 12 weeks [46], and demonstrated an OS benefit of 67 vs. 52 months (*p* = 0.022). However, the control arm is not relevant to the relapse treatment of today. The other trial, the German ReLApsE trial, randomized patients to either Rd continuously or to Rd induction, transplant and lenalidomide maintenance, and showed no difference in progression-free or OS [47]. The panel consensus of both the ESMO and NCCN guidelines still states that salvage transplant may be considered in patients depending on the time from the preceding transplant; for instance, 36 months. It is likely that this should be given as an addition to the best relapse treatment available and not as an alternative.

High dose therapy is associated with significant morbidity, prolonged recovery time in a proportion of patients, and increased risk of secondary malignancies [48]. Samur et al. recently presented evidence of a significant increase in mutational burden in myeloma cells at relapse after high-dose-melphalan [49]. Consequently, as front-line induction treatment continues to improve, new studies will continue to challenge the paradigm of transplant with alternative options, including use of new immunotherapeutic agents, and the use of SCT depending on MRD status.

### 2.3. Current Use of Cyclophosphamide

The alkylating agent cyclophosphamide has been used as a second option to melphalan in the treatment of MM for decades. Cyclophosphamide is known to be less myelotoxic than melphalan and does not require dose adjustment in cases of renal insufficiency [50]; consequently, it may present a favourable safety profile compared to melphalan in frail individuals or patients with an impaired renal function. Cyclophosphamide is reported to have therapeutic effects at a wide range of doses, both by oral and intravenous routes [51,52]. Low doses of oral cyclophosphamide, often 50 mg daily or 100 mg every other day, have been extensively used in combination with corticoids for advanced disease in frail and elderly patients unable to tolerate more toxic combinations. A phase 3 trial compared carfilzomib monotherapy at 27 mg/m^2^ versus corticoids with or without low dose cyclophosphamide in patients who had received at least three prior treatments for MM and were refractory to their most recent therapy [53]. Ninety-five percent of patients in the control arm received the combination of a corticoid and cyclophosphamide. Although the overall response rate was superior for carfilzomib (19.1% vs 11.4%), no differences were found in terms of progression-free or OS (3.3–3.7 and 10 months, respectively).

In the front-line setting, induction treatment with bortezomib, cyclophosphamide and dexamethasone (VCD or CyBorD) appears to be less effective than standard combinations of bortezomib with thalidomide or lenalidomide and dexamethasone [54,55] and does not appear to add to the combination of bortezomib and dexamethasone alone [56]. However, due to the adverse toxicity profile of thalidomide and the unavailability of lenalidomide in certain countries, VCD is still an option in some centers [57,58], and is also a second option recommendation in several guidelines [36,39]. On the other hand, the addition of cyclophosphamide does not appear to add any benefit to the combination of an immunomodulatory drug, bortezomib and dexamethasone as front-line induction [59], or to lenalidomide and dexamethasone in the non-transplant setting [27]. The benefit of the addition of cyclophosphamide to carfilzomib and dexamethasone is also uncertain either in the front-line setting [60] or in relapsed and refractory MM patients [61], although in this later communication, lenalidomide-refractory patients appeared to benefit from the addition of cyclophosphamide to the proteasome inhibitor.

While the alkylating effect is predominant when cyclophosphamide is used at high doses, more recent studies have observed that cyclophosphamide at lower doses, also described as metronomic dosing, has significant immunomodulatory activity [62,63]. Although there is no clear borderline between low and high doses, new roles for the metronomic use of cyclophosphamide are being explored in the context of immune-mediated therapies, particularly immunomodulating agents. Weekly doses of cyclophosphamide can be safely added to the standard doses of lenalidomide and dexamethasone in relapsed/refractory myeloma [64,65]. Either 600 mg cyclophosphamide flat dose on days 1 and 8 [64], or 300 mg/m^2^ on days 1, 8, and 15 [65], of a 28-day cycle, presented a manageable hematologic toxicity, providing overall response rates of 81 and 94%, respectively, and a PFS projected over 2 years and of 16 months, although no large trial has been performed to confirm such findings. More interestingly, the REPEAT study, including 82 patients, was developed to test the hypothesis that the addition of low-dose metronomic oral cyclophosphamide to lenalidomide could induce a synergistic immunomodulatory effect [66]. The final dose of 25 mg lenalidomide on days 1 to 21, combined with continuous cyclophosphamide 50 mg/d and prednisone 20 mg/d, showed an overall response rate of 67% and a median PFS and OS of 12.1 and 29.0 months, respectively, which are promising results considering that all 82 patients in the trial were lenalidomide refractory and 66% were also bortezomib refractory. Such results are comparable to the results of the two previous studies in which a less refractory population was included, and even comparable with the results of cyclophosphamide with lenalidomide and dexamethasone in the front-line setting, although the results have not been reproduced by other groups. The findings of the REPEAT trial may point out major differences in the immune effect of cyclophosphamide at different schedules and doses. The observation that the immunomodulatory properties of lenalidomide can be restored despite a clinical lenalidomide-refractory status can have implications for the development of future combination therapies. The mechanisms of such immunologic effects are far from being elucidated yet [67], but continuous low-dose cyclophosphamide appears to enhance anti-tumor immunity, via the depletion of regulatory T-cells, more efficiently than higher pulsed doses [68].

Similar immune-activating effects are likely to happen in combination with pomalidomide-dexamethasone. The first reported phase 1/2 trial to explore the combination [69] considered pomalidomide up to 2.5 mg/day as the maximum tolerated dose in combination with cyclophosphamide at 50 mg and prednisone at 50 mg every other day. The combination was given for six 28-day cycles, and then patients continued with pomalidomide and prednisone alone. All 69 patients had been exposed to lenalidomide and a third of them were refractory. The overall response rate was 51% (three patients, 5%, achieved a complete response) and the PFS was 10.4 months. Similarly to lenalidomide and cyclophosphamide combinations, toxicity was mainly hematologic but manageable (basically neutropenia, grade 3 or higher, in 52% of patients) and the combination was considered to be safe. Other authors [70,71] have reported data for standard doses of pomalidomide at 4 mg on 21 out of 28 days, and weekly 40 mg doses of dexamethasone, in combination with pulsed cyclophosphamide at a 300 mg or 400 mg dose, with a similar hematological toxicity (20–30% grade 3–4 neutropenia) and a 60 to 76% overall response rate. However, PFS was 7 months only in a mostly lenalidomide refractory population. In all, the results suggest a likely benefit for the addition of cyclophosphamide to pomalidomide alone. The only formal comparison was made in a randomized, multicenter phase 2 trial [72]. Following a dose escalation phase 1 cohort, lenalidomide refractory myeloma patients who had received >2 prior therapies were randomized to receive standard doses of pomalidomide and weekly dexamethasone in 28-day cycles (34 patients) or pomalidomide and dexamethasone with an additional 400 mg cyclophosphamide orally on days 1, 8, and 15 (36 patients). The overall response rate was significantly superior (*p* = 0.035) for the combination with cyclophosphamide (65%) (95% confidence interval [CI], 49–81%) compared to the control arm (39%) (95% CI, 23–55%); median PFS was not significantly different between arms, although it tended to be higher in the cyclophosphamide arm, 9.5 months (95% CI, 4.6–14) versus 4.4 (95% CI, 2.3–5.7). Toxicity was manageable in both arms. The results appear to be only slightly inferior to those reported for the bortezomib, pomalidomide and dexamethasone combination in the OPTIMISMM trial [73] when taking into account that the study included a relevant proportion of patients who were also refractory to proteasome inhibitors (75% refractory to bortezomib and 41% to carfilzomib), who were not included in OPTIMISMM. Weekly oral cyclophosphamide at 300 mg flat dose in addition to standard pomalidomide and dexamethasone was evaluated in less refractory patients in another nonrandomized, multicenter, phase 2 trial [74]. Patients treated front-line with lenalidomide-bortezomib-dexamethasone with or without stem-cell transplantation followed by lenalidomide maintenance were included at first relapse after discontinuation of lenalidomide maintenance; consequently, no patient was lenalidomide refractory. Responses were obtained in 82/97 (85%) patients evaluated, with only 1 (1%) obtaining a complete remission. The 34-month PFS must be interpreted cautiously, taking into consideration that patients who had not received an SCT front-line received SCT after five cycles of pomalidomide, dexamethasone and cyclophosphamide. The addition of cyclophosphamide to patients presenting a suboptimal response to pomalidomide and dexamethasone has also been evaluated in the PERSPECTIVE phase 2 trial [75]. Patients on treatment with standard pomalidomide and dexamethasone, after at least two prior treatment lines including bortezomib and lenalidomide, were considered for the addition of cyclophosphamide 500 mg/m^2^ intravenously on days 1 and 15 if they presented disease progression or achieved less than a partial remission after three treatment cycles. Cyclophosphamide was given for a maximum of 12 cycles. Thirty-six patients were treated; of those progressing on pomalidomide and dexamethasone, only 25% (4 out of 16) obtained a partial or very good response. Of the remaining 20 patients who had presented stable disease or minor response to pomalidomide and dexamethasone alone, 33% and 80%, respectively, upgraded to partial response or better. Only recently, a single-center phase 2 study has reported results for the continuous use of 50 mg cyclophosphamide (days 1–21 of a 28 day cycle) in combination with standard doses of pomalidomide and dexamethasone in 33 lenalidomide refractory patients, 55% of whom were also refractory to a proteasome inhibitor [76]. The overall response rate was 73% and median progression-free and OSs were 13 months and 57 months, respectively. Based on such small trials, the addition of metronomic cyclophosphamide to pomalidomide and dexamethasone appears to improve the results of pomalidomide and dexamethasone alone without much safety compromise, although the overall benefit is difficult to compare with approved alternatives, particularly bortezomib, pomalidomide and dexamethasone. On the other hand, in certain settings, the combination may be cost-effective in comparison with the newly approved combinations of pomalidomide and dexamethasone with monoclonal antibodies [77,78,79].

Low-dose cyclophosphamide is a well-tolerated oral option, which may be useful in heavily treated older patients, when the therapeutic objective is restricted to symptomatic control and preservation of quality of life. Within modern myeloma therapy, the role of metronomic cyclophosphamide is likely to develop further despite the mechanisms underpinning its apparent synergism with immunomodulatory agents remaining insufficiently characterized, and its benefit in combination with monoclonal antibodies and new immunotherapy including cellular therapies still being unknown. A phase 2 trial evaluating daratumumab with cyclophosphamide, dexamethasone and pomalidomide is already ongoing [80]. Several important considerations, including best cyclophosphamide dosing and schedule, optimal partners, and best settings, remain to be elucidated.

### 2.4. Current Use of Bendamustine

The history of bendamustine is unusual. The drug was developed in East Germany behind the iron curtain, and was used to treat both lymphoma, lung cancer and myeloma in the 1970s, alas with few published and validated studies. Biochemically, it has some additional features compared to other alkylators, including a benzimidazole ring with antimetabolite-like properties [81]. It does cause DNA breaks like other alkylators [82], but, in addition, activates a base excision DNA repair pathway which is not seen with other drugs such as cyclophosphamide and melphalan. While other alkylating agents show a large degree of sensitivity overlap when tested in tumor lines, bendamustine displays unique non-cross-resistant features [83]. Bendamustine has established indications in the treatment of Cold Agglutinin Disease [84], Waldenströms macroglobulinemia [85], Hairy Cell Leukemia [86], Chronic Lymphocytic Leukemia [87], Mantle Cell Lymphoma [88], Follicular Lymphoma [89] and Marginal Zone Lymphoma [90]. It is also on the WHO list of essential medicines (https://apps.who.int/iris/handle/10665/325771, accessed on 18 May 2021). However, due to the scarcity of phase-3 trials and a very crowded field of new treatments, its use in MM is marginal. The new ESMO guidelines do not mention bendamustine as an option for relapsed/refractory myeloma [36].

The pivotal trial for bendamustine in myeloma, which led to the still existing marketing approval in newly diagnosed myeloma, was a randomized trial comparing Bendamustine-prednisolone (BP) with the standard-of-care at that time, melphalan-prednisolone (MP) [91]. BP demonstrated a higher rate of complete response (32 vs. 13%; *p* = 0.007), a longer time to treatment failure, a later primary endpoint of the study (14 vs. 10 months; *p* < 0.02), and a significantly improved quality-of-life in several dimensions. This is the only example in myeloma where one alkylating agent proved superior to another in a head-to-head trial, although OS did not differ. In the same period, MP was improved by the addition of bortezomib or thalidomide (see Section 2.1), which both increased OS compared to MP alone. Since both bortezomib and thalidomide had significant neuropathy issues, BP was approved for newly diagnosed myeloma with neuropathy, precluding the use of bortezomib and thalidomide, and this is still the only approval for bendamustine in myeloma.

No randomized phase-3-trials with bendamustine have been performed in the relapsed/refractory myeloma setting. The lack of randomized trials is partly caused by prolonged legal discussions of patents and ownership. However, the belief in the drug’s efficacy has led to a number of phase-1 and phase-2 trials (Table 1) that have combined bendamustine with other established active agents against myeloma. With the large gap that exists between approval of regimens and national reimbursement of the novel regimens in most countries, the fact that bendamustine is now generic creates the opportunity to use these combinations when other combinations are not available.

As seen in Table 1, most combination studies in RRMM have been performed with thalidomide, lenalidomide or bortezomib. Today, lenalidomide has replaced thalidomide, and both bortezomib and lenalidomide are most commonly used in first line treatment, making combinations with these drugs less useful at relapse. Carfilzomib and pomalidomide are becoming the backbone of triplet combinations in the relapsed and refractory setting. Two triplet combinations with carfilzomib are approved, in combination with lenalidomide-dex [109], which is less of an option as more patients are refractory to lenalidomide at relapse, as well as the newly approved combination with daratumumab [110]. Pomalidomide also has an approved combination with a first-line drug, bortezomib [73], but it is also approved with the monoclonal antibodies elotuzumab [79] and isatuximab [78]. Reimbursement issues make the use of these two triplet regimens difficult in many countries. In such cases, the use of triplets with the generic drug bendamustine may be considered as long as they prove efficacious and safe in phase 2 trials.

The combination with pomalidomide-dexamethasone was tested in a phase I/II dose escalation trial in patients who were pomalidomide naïve, refractory to full-dose lenalidomide and refractory to the last treatment [105]. The study included 38 patients, who, in addition to lenalidomide, had all received bortezomib, 82% ASCT and 32% carfilzomib. The median previous number of lines was 5 (3–8). Treatment with bendamustine was given at 120 mg/m^2^ at day 1 of 28, for 12 cycles, Pd was continued until progression. The MTD was defined to be pomalidomide 3 mg (days 1–21) and dexamethasone (40 mg weekly). Hematologic toxicities were prevalent but could be effectively managed; the majority of patients discontinued treatment due to disease progression or lack of response. Overall response rate was 61% and median PFS was 9.6 months, comparing very favorably to the pivotal MM-003 study (31% and 4.0 months; [111]), with a similar patient population.

In another study, bendamustine was combined with carfilzomib and dexamethasone in a phase I/II trial in patients with at least two previous lines of treatment [106]. The study included 63 patients, where 75% had received ASCT, 87% bortezomib, and 86% immunomodulatory drugs. The median previous number of lines was 4 (2–9). Treatment with bendamustine was given at 70 mg/m^2^ at day 1 and 8 of 28 for 8 cycles, before carfilzomib-dex maintenance every 14 days was continued until progression. The MTD was defined to be carfilzomib (27 mg/m^2^) (two first days with 20 mg/m^2^) administered on days 1, 2, 8, 9, 15 and 16, with dexamethasone (20 mg) administered on the same days plus days 22 and 23. In addition, in this study the hematological toxicity was manageable, and there were no new safety signals. The overall response rate was 51% and the median PFS was 11.6 months, with the PFS being higher than the Kd doublet with 56 mg/m^2^ in a slightly more treated population (55%, 4.1 months) [112].

In situations where triplets are not an option, bendamustine can also be combined with dexamethasone alone. Data are not extensive, but in an Italian study of eight frail patients with a median of four previous lines, the patients were treated with bendamustine (60 mg) on days 1, 8 and 15, and dexamethasone (20 mg) weekly [113]. All patients had been treated with bortezomib (75% refractory) and lenalidomide (62.5% refractory), but with a median age of 76, only one of the patients had received a transplant. The combination was safe and provided an overall response rate of 55% and a PFS of 9.1 months.

Bendamustine has also been tested in newly diagnosed myeloma as part of conditioning before autologous stem cell transplant, and in multi-drug combinations. Two studies have evaluated the addition of bendamustine to melphalan as conditioning before the first ASCT [114,115], and Martino et al. tested this combination as part of the second transplant in a tandem strategy [116]. Combinations with bortezomib-dexamethasone have also been tried in patients with newly diagnosed myeloma with or without transplant [117,118]. Bendamustine has also been explored in combination with carfilzomib in this population [119]. Bendamustine in combination with bortezomib or carfilzomib in the front-line setting does not appear to provide relevant improvements over already established combinations. Other regimens including lenalidomide and daratumumab are probably better for induction in newly diagnosed patients, and the potential benefit of adding bendamustine to melphalan as conditioning would need to be confirmed in phase-3 studies.

In summary, bendamustine is a unique drug with non-overlapping resistance to other alkylators. Evidence for its benefit is hampered by a lack of phase-3 trials, but it remains an alternative option if other triplets with carfilzomib and pomalidomide are not available.

## 3. Melflufen

Melphalan flufenamide (melflufen) is a first-in-class peptide-drug conjugate that targets aminopeptidases and rapidly releases alkylating agents into tumor cells [120,121,122,123]. Melflufen is rapidly taken up by myeloma cells due to its high lipophilicity [120,124]. The drug is dependent on its immediate cleavage by peptidases, which unleashes its more hydrophilic alkylator payload that remains entrapped within the cell (Figure 2).

Peptidases are highly present in MM cells’ cytoplasm [125,126,127]. Once inside the myeloma cell, melflufen rapidly induces irreversible DNA damage and apoptosis via an alkylator-like mechanism of cytotoxicity that is independent of p53 [124,128,129]. Melflufen is 50-fold more potent than melphalan in myeloma cells in vitro because the combined effects of its high lipophilicity and intracellular binding to aminopeptidases results in increased intracellular alkylator effect [120,121,122,123].

A Phase I/II study O-12-M1 [121] established an optimal cycle length of 28 days and a maximum tolerated flat dose of melflufen of 40 mg administered as a single intravenous dose at day 1 of the cycle. Patients in the trial had received a median of four prior lines of therapy and 31% achieved at least a partial response. After a median follow-up of 28 months, the median PFS, duration of response (DOR) and OS were 5.7, 8.4 months and 20.7 months, respectively. Toxicity was basically hematological, including thrombocytopenia (73%), neutropenia (69%) and anemia (64%). Pneumonia was the most common infectious complication (11%). The HORIZON (OP-106) subsequent phase II trial was pivotal for establishing the efficacy of the drug [121]. The trial included 154 heavily pretreated and poor-risk patients with RRMM refractory to pomalidomide, an anti-CD38 mAb, or both, who had exhausted most salvage therapy options. Patients had a median age of 64 years (range, 35–86), and 32% had ISS stage III disease, 38% had high-risk cytogenetics, and 32% had extramedullary disease. They had received a median of five prior lines of therapy (range, 2–12), 83% had received prior alkylator therapy, 71% had triple-class refractory disease, and 69% had received ≥1 prior autologous stem cell transplant. Among 125 efficacy-evaluable patients, the ORR was 29% (including 1 stringent complete response and 12 VGPR), thus confirming the efficacy observed in initial studies. The median PFS was 4.2 months and the median OS was 11.6 months; DOR was 5.5 months. Triple-class refractory patients and individuals with extramedullary disease achieved similar results to the overall population with an overall response rate of 24%, a median duration of response of 7.5 and 5.1 months, respectively, and a median OS of 11.3 and 8.1 months, respectively. Melflufen and dexamethasone were generally safe and well tolerated. In the overall 154 safety-evaluable individuals, 97% of patients experienced a treatment-emergent AE and 85% experienced a grade 3 or 4 treatment-emergent AE, but the incidence of nonhematologic AEs was low. Common grade 3/4 toxicity consisted mostly of thrombocytopenia (25%/51%), neutropenia (32%/47%), and, less frequently, anemia (42%/<1%). These encouraging results have triggered an ongoing registrational phase III trial comparing melflufen and dexamethasone with pomalidomide dexamethasone (OP-103) [130], which completed the recruitment of 450 individuals in 2020. The safety profile and mechanism of action of melflufen also suggest potentially active combinations. An ongoing phase I/II trial, ANCHOR (OP-104) [131], is currently evaluating the safety and efficacy of melflufen and dexamethasone in combination with daratumumab or bortezomib in patients who had received 1 to 4 prior lines of therapy. As recently communicated, 13 patients with a median of 3 prior lines of therapy, of whom 95% had been exposed to a proteasome inhibitor (range, 2–4), were treated with melflufen (six with 30 mg; seven with 40 mg) plus dexamethasone and bortezomib, with a median treatment duration of 9 months (range 1.4 to 29). ORR was 62%, four patients achieved a VGPR and one a CR. In the other treatment arm, 33 patients, all of them lenalidomide and proteasome inhibitor refractory but not exposed to daratumumab, and with a median of 2 previous lines of therapy, received melflufen (6 with a 30-mg dose; 27 with a 40-mg dose) in combination with daratumumab and dexamethasone. The ORR (≥PR) was 73% (including 2 CR and 10 VGPR), and the median PFS was 13 months. No dose-limiting toxicities were observed in any cohort and adverse events were consistent with those observed with melflufen and dexamethasone and those already known for daratumumab and bortezomib. No relevant treatment-related adverse events were observed other than hematological toxicity. Despite being an alkylating agent, melflufen does not appear to cause alopecia and the incidence of mucositis is low because of the selectivity of alkylation in cells with high expression of aminopeptidases.

In summary, melflufen has demonstrated durable responses in triple-refractory MM patients and a manageable safety profile. Treatment-related adverse events associated with melflufen and dexamethasone are primarily hematologic and rates of nonhematologic AEs are low. Based on this encouraging activity and acceptable safety profile, combination with proteasome inhibitors and anti-CD38 monoclonal antibodies may offer interesting results in less advanced patients. Melflufen was recently approved by the FDA.

## 4. Current Standard of Care

Today, the status of HDM-ASCT is still strong and is the recommended/preferred option in first-line transplant-eligible patients [36,39]. The role of tandem transplant is more controversial but is recommended for high-risk patients, for patients after treatment with VCD induction, and is an option for all patients. It is our opinion that salvage transplant should not be used as an alternative to the approved relapse triplets, but could be considered after a limited number of cycles of triplet therapy and always followed by continuous therapy.

In newly diagnosed transplant-ineligible patients, Dara-VMP is an adequate option if the lenalidomide-containing, alkylator-free triplets (VRd and Dara-Rd) [132,133] are not available.

Alkylator regimens at time of relapse are only supported by phase-2 single arm trials, contrary to the randomized phase-3 trials supporting carfilzomib- and pomalidomide-based triplets. Carfilzomib or pomalidomide, combined with an anti-CD38 antibody, have shown clear superiority over the doublets with dexamethasone [78,110]. In situations where the documented triplets are not available, we advocate considering the addition of cyclophosphamide or bendamustine, especially in patients with good bone marrow function and limited previous exposure to alkylators.

In late line disease, several new options have emerged and recently been approved by the FDA and/or EMA. Amongst these is the alkylator melflufen, but also the nuclear transport inhibitor selinexor and the antibody-drug-conjugate drug belantamab mafodotin. The alkylator melflufen has the longest median PFS amongst these, but the differences are small, and the populations not entirely the same. As of today, they are all valid options, and the choice between them should be made based on availability, comorbidities, and patient choice. The most important adverse events are cytopenias for melflufen; cytopenias, fatigue and diarrhea for selinexor; and corneal events for belantamab. More information on these drugs in combination with other agents will be available in the near future.

Whether some clinical features, such as highly proliferative or extramedullary disease (EMD), should increase the inclination to administer alkylators is, as yet, unknown, although melflufen performed well in EMD patients [134]. Although some preclinical studies suggest biomarkers affecting response to alkylators [135], these data are so far too immature to impact treatment choice.

## 5. Future Use of Alkylators

Even though alkylators have been a mainstay in myeloma treatment for decades, their use is continuously being challenged due to their potential long-term toxicity. Two recent randomized studies compared transplant with a non-transplant approach in first-line treatment, but, in both cases, the transplant arm prevailed [37,43]. Still, the reduced quality-of-life in the period after transplant, as well as the increased mutational burden and increase in secondary malignancies seen after alkylator use, remains a concern. Efficacious immunotherapy, including chimeric-antibody-receptor T cells (CAR-Ts) and bispecific antibodies, have provided better results in terms of response rates and progression-free survival in later lines, and the possibility of including these agents as part of first-line treatments may again challenge HDM in first line [136,137]. Currently, several trials are being performed in which these agents are incorporated in first-line treatment in a design without HDM (e.g., NCT04196491, ClinicalTrials.gov).

Alkylating agents, as part of front-line induction treatment in transplant-ineligible patients, have been replaced by more active combinations, including both proteasome inhibitors, immunomodulating agents and monoclonal CD38 antibodies. Still, as long as most patients are not cured, the exposure to alkylators at some point throughout the disease will continue to be beneficial. The lack of comparative trials between alkylators makes it difficult to state with certainty that there are efficacy differences between the different drugs, but with the new delivery mechanism and the recent phase-2 data, the novel alkylator melflufen is the most promising agent today. Phase 3 combination studies are underway, and will shed light on the benefits of melflufen in combination, first with pomalidomide [130], and next with daratumumab (NCT04649060, ClinicalTrials.gov).

Currently, many studies are being performed for patients with smouldering multiple myeloma. As of today, we do not see a place for alkylators in this setting. The concern regarding increased mutational burden is higher in these patients, and the aim should be to cure them without inflicting genetic damage.

## 6. Conclusions

For decades, alkylating agents, together with corticosteroids, have been the only efficacious drugs for the treatment of MM and, to date, they remain a cornerstone of MM treatment. Despite rising concern regarding induced DNA damage caused by this class of drugs, most patients with MM, in all parts of the world, will receive an alkylating agent in their disease course. The use of high-dose melphalan with stem cell transplant is still increasing in all global regions, and new alkylator-containing regimens have recently been approved by the FDA and the EMA.

In addition, a novel alkylator is now about to enter the market. Melflufen has a distinct mechanism-of-action and becomes active only in peptidase-rich cells, leading to less off-target effects and higher potency towards myeloma cells. Phase 1 and 2-studies have been published, and results from the first phase-3 trial are eagerly awaited.

## Figures and Tables

**Figure 1 cancers-13-02465-f001:**
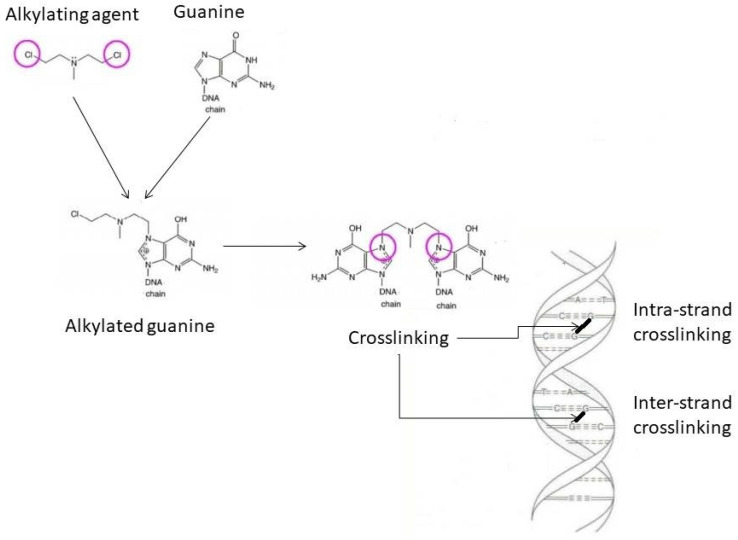
Mechanism of action of alkylating agents. Alkylation of guanine may cause the excision or the opening of the guanine ring, thus interfering in base repairing. Bifunctional alkylating agents also cause intra and inter-strand crosslinking. Cross-linking interferes both with transcription and replication.

**Figure 2 cancers-13-02465-f002:**
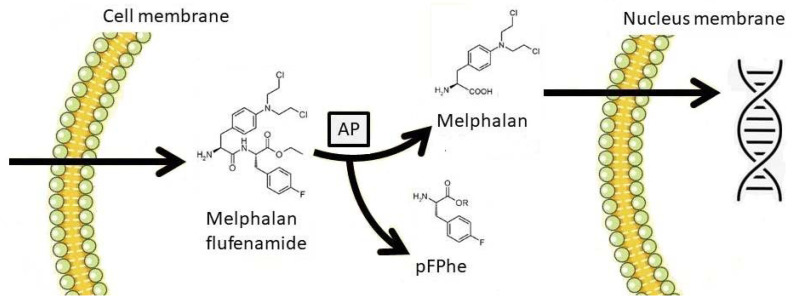
Melphalan flufenamide (melflufen) diffuses rapidly through the cellular membrane due to its lipophilic properties. Cytoplasmic aminopeptidases (AP) cleave melphalan from its peptide carrier (pFPhe). The hydrophilic melphalan residue is trapped into the cell and exerts its alkylating effect in the DNA.

**Table 1 cancers-13-02465-t001:** Combination studies using bendamustine in the relapsed/refractory multiple myeloma setting.

Combination	Previous Lines (Median)	*N*	Dose of Study (a)	ORR	PFS	Reference
Thalidomide-Prednisolone	1	28	60 mg/m^2^ day 1, 8, 15 (of 28)	86%	11 m	Pönisch BJH 2008 [92]
Thalidomide-Dexamethasone	1 (b)	9	120 mg day 1 (of 28)	55%	NR	Ramasamy BJH 2011 [93]
Lenalidomide-Dexamethasone	3	29	75 mg/m^2^ day 1, 2 (of 28)	52%	6.1 m	Lentzsch Blood 2012 [94]
Bortezomib	6	40	90 mg/m^2^ day 1, 4 (of 28)	52%	8.4 m	Berenson BJH 2013 [95]
Lenalidomide-Dexamethasone	2	21	75 mg/m^2^ day 1, 2 (of 28)	76%	48% at 18 m	Pönisch BJH 2013 [96]
Bortezomib-Dexamethasone	3 (c)	36	60 mg/m^2^ day 1, 2 (of 21)	67%	10 m (eGFR 15–59)	Pönisch JCancResClinOnc 2013 [97]
Bortezomib-Dexamethasone	2	79	70 mg/m^2^ day 1, 4 (of 28)	61%	9.7 m	Ludwig Blood 2014 [98]
Bortezomib-Dexamethasone	1 (d)	73	70 mg/m^2^ day 1, 8 (of 28)	58%	10.8 m	Rodon Haematologica 2015 [99]
Thalidomide-Dexamethasone	3	94	60 mg/m^2^ day 1, 8 (of 28)	46%	7.5 m	Schey BJH 2015 [100]
Lenalidomide-Dexamethasone	3		75 mg/m^2^ day 1, 2 (of 28)	49%	11.8 m	Kumar AJH 2015 [101]
Lenalidomide-Dexamethasone	2	38	40 mg/m^2^ day 1, 2 (of 28)	48% (phase 2)	10 m	Pozzi LeukLymph 2017 [102]
Lenalidomide-Dexamethasone	1	50	75 mg/m^2^ day 1, 2 (of 28)	89%	18.6 m	Mey BJH 2017 [103]
Lenalidomide-Dexamethasone	1	25	75 mg/m^2^ day 1, 2 (of 28)	88%	22 m	Beck JCancResClinOnc 2017 [104]
Pomalidomide-Dexamethasone	5	38	120 mg/m^2^ day 1 (of 28)	61%	9.6 m	Sivaraj BCJ 2018 [105]
Carfilzomib-Dexamethasone	4	63	70 mg/m^2^ day 1, 8 (of 28)	51%	11.6 m	Gramatzki ASCO2018 Abstract 8019 [106]
Thalidomide-Dexamethasone	3.5	30	60 mg/m^2^ day 1, 8, 15 (of 28)	37%	6.2 m (TTF)	Mian BJH 2019 [107]
Ixazomib-Dexamethasone	4	28	80 mg/m^2^ day 1, 2 (of 28)	61% (phase 2)	5.2 m	Dhakal BCJ 2019 [108]

(a) MTD in dose escalation trials. (b) All patients in ESRD. (c) All patients eGFR < 60 mL/min. (d) All patients > 65 years of age.

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
