# Peer review of "Current and Novel Alkylators in Multiple Myeloma"

_cancers, 2021, doi:10.3390/cancers13102465_

Round 1

Reviewer 1 Report

In this review, Authors focus on the therapeutic use of alkylators in multiple myeloma.

The review is well written and organised discussing about the old and current status of alkylating drugs in the standard-of care in myeloma patients.

Author Response

Thank you for this feedback.

Reviewer 2 Report

This is a well written and comprehensive review of the role of alkylating agent chemotherapies in multiple myeloma (MM). As the authors note, alkylating agents were the first effective chemotherapy for MM and remain a mainstay of therapy today. The authors also note that their focus was “non-systematized” and it appears that the focus here is on compiling the extensive data on the older alkylating agent bendamustine in combinations with other agents, and a shorter review about what is known about the newer alkylating prodrug melflufen. The authors spend a good deal of space describing the data and references on bendamustine combinations in MM, even though they also note that, “current use in MM is marginal,” and that “new ESMO guidelines, do not mention bendamustine as an option for relapsed/refractory myeloma.” Alkylating agents are have been used in all stages of myeloma treatment, including induction, chemo-mobilization, transplant conditioning, and in relapsed/refractory RRMM. Additional settings of alkylator use might be included for completeness:

  1. Autologous transplantation continues to provide survival benefit in the era of novel therapies. However, additional effective immunotherapies (BCMA-targeted CAR-T and bispecific antibodies) have shown strong results in RRMM and will be studied earlier in disease. What do the authors think of the “alternate hypothesis” that alkylating agents will not remain a mainstay but will only be used in the RRMM setting in the era of immunotherapy?
  2. I was disappointed that the authors did not provide any guidance or speculation in the conclusions how they feel optimal treatment strategies should use alkylating agents going forward? From all the collected data, can the authors hypothesize?
  3. Arkansas’s use of alkylating agent heavy VDT-PACE chemo-mobilization has not to my knowledge gained wide-spread acceptance, but given Dr. Barlogie’s contributions to the field, some may argue that Total Therapy style chemo-mobilization should be discussed.
  4. The authors note that alkylators in induction chemo e.g. CyBorD (VCD) is being eclipsed by the use of combinations that incorporate newer agents such as Kyprolis and Daratumumab/anti-CD38 antibodies up front. Therefore, role of alkylating agents appears likely to remain as HD melphalan in transplant, and in regimens for relapsed or refractory disease. It is unclear what the authors believe the role of alkylating agents will be in the future as novel effective therapies continue to be developed.
  5. There are little to no data that one alkylating agent any more benefit than any other. Lack of randomized trials is disheartening. Most trials industry sponsored trials focus on obtaining regulatory approval for novel drugs, and novel combinations. How do the authors anticipate the MM field going to determine optimal therapies with the current approaches?
  6. There are some data that suggest that the effectiveness of alkylating agents is modulated by the underlying genetics of the disease (e.g. Oliveira et al FASEB Bioadv. 2019 Jul;1(7):404-414.) Do the authors feel that there any role for alkylating agents to be used as “personalized medicine?”
  7. The authors mentioned the risk of secondary cancers only once but might this risk further temper the use of these agents in the era of immunotherapies?

Author Response

Thank you for your feedback. 

To comment 1:

We agree that this is an interesting discussion, which was missing in the first manuscript. We have added a section before the conclusion, called "Future use of alkylators" where we address this. 

To comment 2: 

Thank you for this comment. We have added a section before the section "Future use...", called "Current standard of care" where we address this. 

To comment 3: 

Thank you for this comment. We have mentioned this in the end of section "1".

To comment 4:

Thank you for this comment. This has been addressed in the new section 5. "Future..."

To comment 5,6 and 7:

Thank you for these comments. They have also been addressed in the new section 5

Reviewer 3 Report

This is a comprehensive clinical review on the past and current use of alkylators for the treatment of multiple myeloma. The authors have detailed the history of the use of alkylators and have provided a thorough review of the existing literature.  I have a few important comments for the authors

Please give an overview of what the authors see as the future for using alkylators in myeloma.

What are the alkylator based combinations that the authors think will be used in the future?

Which patients might benefit?

Do the authors see a role for alkylators in delaying progression of high risk smoldering myeloma?

With increased immune therapies being developed, what is the role of alkylating agents in myeloma?

What could be potential biomarkers to identify patients that might benefit from alkylating agents? For instance, highly proliferating disease, particular cytogenetic abnormalities?

Author Response

Thank you for your feedback.

Your comments and questions are valid and we think it has made the manuscript better. 
We have addressed all questions/comments in two new sections (4: Current... and 5: Future....).

Round 2

Reviewer 2 Report

Appreciate the response to comments and believe the manuscript imrpoved with the authors' additions.

Strong work, this will be a good resource.